# Semantic Decomposition of Question and SQL for Text-to-SQL Parsing

**Ben Eyal** and **Amir Bachar** and **Ophir Haroche**

**Moran Mahabi** and **Michael Elhadad**
Dept of Computer Science
Ben Gurion University
Beer Sheva, Israel

## Abstract

Text-to-SQL semantic parsing faces challenges in generalizing to cross-domain and complex queries. Recent research has employed a question decomposition strategy to enhance the parsing of complex SQL queries. However, this strategy encounters two major obstacles: (1) existing datasets lack question decomposition; (2) due to the syntactic complexity of SQL, most complex queries cannot be disentangled into sub-queries that can be readily recomposed.

To address these challenges, we propose a new modular Query Plan Language (QPL) that systematically decomposes SQL queries into simple and regular sub-queries. We develop a translator from SQL to QPL by leveraging analysis of SQL server query optimization plans, and we augment the Spider dataset with QPL programs. Experimental results demonstrate that the modular nature of QPL benefits existing semantic-parsing architectures, and training text-to-QPL parsers is more effective than text-to-SQL parsing for semantically equivalent queries.

The QPL approach offers two additional advantages: (1) QPL programs can be paraphrased as simple questions, which allows us to create a dataset of (complex question, decomposed questions). Training on this dataset, we obtain a Question Decomposer for data retrieval that is sensitive to database schemas. (2) QPL is more accessible to non-experts for complex queries, leading to more interpretable output from the semantic parser.

## 1 Introduction

Querying and exploring complex relational data stores necessitate programming skills and domain-specific knowledge of the data. Text-to-SQL semantic parsing allows non-expert programmers to formulate questions in natural language, convert the questions into SQL, and inspect the execution results. While recent progress has been remarkable on this task, general cross-domain text-to-SQL models still face challenges on complex schemas and queries. State of the art text-to-SQL models show performance above 90% for easy queries, but fall to about 50% on complex ones (see Table 1). This accuracy drop is particularly bothersome for non-experts, because they also find it difficult to verify whether a complex SQL query corresponds to the intent behind the question they asked. In a user study we performed, we found that software engineers who are not experts in SQL fail to determine whether a complex SQL query corresponds to a question in about 66% of the cases (see Table 4). The risk of text-to-code models producing incorrect results with confidence is thus acute: complex SQL queries non-aligned with users intent will be hard to detect.

In this paper, we address the challenge of dealing with complex data retrieval questions through a compositional approach. Based on the success of the question decomposition approach for multi-hop question answering, recent work in semantic parsing has also investigated ways to deal with complex SQL queries with a Question Decomposition (QD) strategy. In another direction, previous attempts have focused on splitting complex SQL queries into spans (e.g., aggregation operators, join criteria, column selection) and generating each span separately.

In our approach, we start from a semantic analysis of the SQL query. We introduce a new intermediary language, which we call Query Plan Language (QPL) that is modular and decomposable. QPL can be directly executed on SQL databases through direct translation to modular SQL Common Table Expressions (CTEs). We design QPL to be both easier to learn with modern neural architectures than SQL and easier to interpret by non-experts. The overall approach is illustrated in Fig. 1. We develop an automatic translation method from SQL to QPL. On the basis of the modular QPL program, we also learn how to generate a natural language *decomposition* of the original question.

**Question**

```
What is the official language spoken in the country whose head of state
is Beatrix?
```

**Gold SQL**

```sql
SELECT T2.Language
FROM country AS T1
JOIN countrylanguage AS T2 ON T1.Code  =  T2.CountryCode
WHERE T1.HeadOfState  =  'Beatrix' AND T2.IsOfficial  =  'T'
```

**Gold QPL**

```
#1 = Scan Table [country] Predicate [HeadOfState = 'Beatrix']
     Output [Code, HeadOfState]
#2 = Scan Table [countrylanguage] Output [CountryCode, Language, IsOfficial]
#3 = Filter [#2] Predicate [IsOfficial = 'T'] Output [CountryCode, Language]
#4 = Join [#1, #3] Predicate [#3.CountryCode = #1.Code] Output [#3.Language]
```

**Computed Question Decomposition (QD)**

```
#1 = Scan the table country and retrieve the code and
     head of state of the country whose head of state is Beatrix
#2 = Scan the table countrylanguage and retrieve the country codes,
     languages and if they're official
#3 = Filter from #2 all the official languages and
     retrieve the country codes and languages
#4 = Join #1 and #3 based on the matching country codes and retrieve
     the language spoken in the country whose head of state is Beatrix
```

**Predicted QDMR**

```
#1 = return countries whose head of state is beatrix ;
#2 = return the official language spoken in the official language of #1
```

Figure 1: Example QPL and Question Decomposition compared to the original SQL query from *Spider* and to the predicted QDMR question decomposition from (Wolfson et al., 2020).

In contrast to generic QD methods such as QDMR (Wolfson et al., 2020), our decomposition takes into account the database schema which is referenced by the question and the semantics of the QPL operations.

Previous research in semantic parsing has shown that the choice of the target language impacts a model's ability to learn to parse text into an accurate semantic representation. For instance, Guo et al. (2020) compared the performance of various architectures on three question-answering datasets with targets converted to Prolog, Lambda Calculus, FunQL, and SQL. They discovered that the same architectures produce lower accuracy (up to a 10% difference) when generating SQL, indicating that

SQL is a challenging target language for neural models. The search for a target language that is easier to learn has been pursued in Text-to-SQL as well (Yu et al., 2018a; Guo et al., 2019; Gan et al., 2021). We can view QPL as another candidate intermediary language, which in contrast to previous attempts, does not rely on a syntactic analysis of the SQL queries but rather on a semantic transformation into a simpler, more regular query language.

In the rest of the paper, we review recent work in text-to-SQL models that investigates intermediary representations and question decomposition. We then present the Query Plan Language (QPL) we have designed and the conversion procedure

we have implemented to translate the existing large-scale *Spider* dataset into QPL. We then describe how we exploit the semantic transformation of SQL to QPL to derive a dataset of schema-dependent Question Decompositions. We finally present strategies that exploit the compositional nature of QPL to train models capable of predicting complex QPL query plans from natural language questions, and to decompose questions into data-retrieval oriented decompositions.

We investigate two main research questions: **(RQ1)** Is it easier to learn Text-to-QPL – a modular, decomposable query language – than to learn Text-to-SQL using Language Model based architectures; **(RQ2)** Can non-expert users interpret QPL outputs in an easier manner than they can for complex SQL queries.

Our main contributions are (1) the definition of the QPL language together with automatic translation from SQL to QPL and execution of QPL on standard SQL servers; (2) the construction of the Spider-QPL dataset which enriches the Spider samples with validated QPL programs for Spider's dataset together with Question Decompositions based on the QPL structure; (3) Text-to-QPL models to predict QPL from a (Schema + Question) input that are competitive with state of the art Text-to-SQL models and perform better on complex queries; (4) a user experiment validating that non-expert users can detect incorrect complex queries better on QPL than on SQL.[1]

## 2 Previous Work

Text-to-SQL parsing consists of mapping a question $Q = (x_1, \ldots, x_n)$ and a database schema $S = [table_1(col_1^1 \ldots col_{c_1}^1), \ldots, table_T(col_1^T \ldots col_{c_T}^T)]$ into a valid SQL query $Y = (y_1, \ldots, y_q)$. Performance metrics include exact match (where the predicted query is compared to the expected one according to the overall SQL structure and within each field token by token) and execution match (where the predicted query is executed on a database and results are compared).

Several large Text-to-SQL datasets have been created, some with single schema (Wang et al., 2020b), some with simple queries (Zhong et al., 2017). Notably, the *Spider* dataset (Yu et al., 2018b) encompasses over 200 database schemas with over 5K complex queries and 10K questions.

It is employed to assess the generalization capabilities of text-to-SQL models to unseen schemas on complex queries. Recent datasets have increased the scale to more samples and more domains (Lan et al., 2023; Li et al., 2023). In this paper, we focus on the *Spider* dataset for our experiments, as it enables comparison with many previous methods.

### 2.1 Architectures for Text-to-SQL

Since the work of Dong and Lapata (2016), leading text-to-SQL models have adopted attention-based sequence to sequence architectures, translating the question and schema into a well-formed SQL query. Pre-trained transformer models have improved performance as in many other NLP tasks, starting with BERT-based models (Hwang et al., 2019; Lin et al., 2020) and up to larger LLMs, such as T5 (Raffel et al., 2020) in (Scholak et al., 2021), OpenAI *CodeX* (Chen et al., 2021) and GPT variants in (Rajkumar et al., 2022; Liu and Tan, 2023; Pourreza and Rafiei, 2023).

In addition to pre-trained transformer models, several task-specific improvements have been introduced: the encoding of the schema can be improved through effective representation learning Bogin et al. (2019), and the attention mechanism of the sequence-to-sequence model can be fine-tuned Wang et al. (2020a). On the decoding side, techniques that incorporate the syntactic structure of the SQL output have been proposed.

To make sure that models generate a sequence of tokens that obey SQL syntax, different approaches have been proposed: in (Yin and Neubig, 2017), instead of generating a sequence of tokens, code-oriented models generate the abstract syntax tree (AST) of expressions of the target program. Scholak et al. (2021) defined the *constrained decoding method* with PICARD. PICARD is an independent module on top of a text-to-text auto-regressive model that uses an incremental parser to constrain the generated output to adhere to the target SQL grammar. Not only does this eliminate almost entirely invalid SQL queries, but the parser is also schema-aware, thus reducing the number of semantically incorrect queries, e.g., selecting a non-existent column from a specific table. We have adopted constrained decoding in our approach, by designing an incremental parser for QPL, and enforcing the generation of syntactically valid plans.

---

[1]All of the datasets and code are available on `https://github.com/bgunlp/qpl`.

| Difficulty | NatSQL + RAT-SQL | Din-SQL GPT-4 | GPT3.5-turbo |
|------------|------------------|---------------|--------------|
| Easy | **91.6%** | 91.1% | 87.7% |
| Medium | 75.2% | **79.8%** | 75.1% |
| Hard | 65.5% | 64.9% | **72.5%** |
| Extra Hard | 51.8% | 43.4% | **53.9%** |
| Overall | 73.7% | 74.2% | **74.3%** |

Table 1: *Spider* Development Set baseline execution accuracy by difficulty level

## 2.2 Zero-shot and Few-shot LLM Methods

With recent LLM progress, the multi-task capabilities of LLMs have been tested on text-to-SQL. In zero-shot mode, a task-specific prompt is prefixed to a textual encoding of the schema and the question, and the LLM outputs an SQL query. Rajkumar et al. (2022); Liu et al. (2023), showed that OpenAI Codex achieves 67% execution accuracy. In our own evaluation, GPT-4 (as of May 2023) achieves about 74% execution accuracy under the same zero-shot prompting conditions.

Few-shot LLM prompting strategies have also been investigated: example selection strategies are reviewed in (Guo et al., 2023; Nan et al., 2023) and report about 85% execution accuracy when tested on Spider dev or 7K examples from the Spider training set. Pourreza and Rafiei (2023); Liu and Tan (2023) are top performers on *Spider* with the GPT4-based DIN-SQL. They use multi-step prompting strategies with query decomposition.

Few-shot LLM prompting methods close the gap and even outperform specialized Text-to-SQL models with about 85% execution match vs. 80% for 3B parameters specialized models on the *Spider* test set, without requiring any fine-tuning or training. In this paper, we focus on the hardest cases of queries, which remain challenging both in SQL and in QPL (with execution accuracy at about 60% in the best cases). We also note that OpenAI-based models are problematic as baselines, since they cannot be reproduced reliably.[2]

## 2.3 Intermediary Target Representations

Most text-to-SQL systems suffer from a severe drop in performance for complex queries, as reported for example in DIN-SQL results where the drop in execution accuracy between simple queries and hard queries is from about 85% to 55% (see also (Lee, 2019)). We demonstrate this drop in

Table 1 which shows execution accuracy of leading baseline models (Gan et al., 2021; Pourreza and Rafiei, 2023) per Spider difficulty level on the development set. The GPT3.5-turbo results correspond to our own experiment using zero-shot prompt. Other methods have been used to demonstrate that current sequence to sequence methods suffer at *compositional generalization*, that is, systems trained on simple queries, fail to generate complex queries, even though they know how to generate the components of the complex query. This weakness is diagnosed by using challenging *compositional splits* (Keysers et al., 2019; Shaw et al., 2021; Gan et al., 2022) over the training data.

One of the reasons for such failure to generalize to complex queries relates to the gap between the syntactic structure of natural language questions and the target SQL queries. This has motivated a thread of work attempting to generate simplified or more generalizable logical forms than executable SQL queries. These attempts are motivated by empirical results on other semantic parsing formalisms that showed that adequate syntax of the logical form can make learning more successful (Guo et al., 2020; Herzig and Berant, 2021).

Most notable attempts include SyntaxSQLNet (Yu et al., 2018a), SemQL (Guo et al., 2019) and NatSQL (Gan et al., 2021). *NatSQL* aims at reducing the gap between questions and queries. It introduces a simplified syntax for SQL from which the original SQL can be recovered. Figure 2 illustrates how this simplified syntax is aligned with spans of the question.

Our work is directly related to this thread. Our approach in designing *QPL* is different from NatSQL, in that we do not follow SQL syntax nor attempt to mimic the syntax of natural language. Instead, we apply a semantic transformation on the SQL query, and obtain a compositional regular query language, where all the nodes are simple executable operators which feed into other nodes in

---

[2]It is most likely that the Spider dataset was part of the training material processed by GPT-x models.

**Question**

```
What type of pet is the youngest animal,
and how much does it weigh?
```

**SQL**

```
SELECT PetType , Weight FROM Pets
ORDER BY Pet_Age LIMIT 1
```

**Spider-SS Decomposition**
*SubSentence:* `What type of pet`
*NatSQL:* `SELECT Pets.Pettype`
*SubSentence:* `is the youngest animal`
*NatSQL:* `ORDER BY Pets.Pet_Age LIMIT 1`
*SubSentence:* `and how much does it weigh?`
*NatSQL:* `SELECT Pets.Weight`

Figure 2: NatSQL and Question Decomposition in *Spider-SS* (Gan et al., 2022)

a data-flow graph according to the execution plan of the SQL query. Our method does not aim at simplifying the mapping of a single question to a whole query, but instead at decomposing a question into a tree of simpler questions, which can then be mapped to simple queries. The design of QPL vs. SQL adopts the same objectives as those defined in KoPL vs. SparQL in (Cao et al., 2022) in the setting of QA over Knowledge Graphs.

## 2.4 Question Decomposition Approaches

Our approach is also inspired by work attempting to solve complex QA and semantic parsing using a *question decomposition strategy* (Perez et al., 2020; Fu et al., 2021; Saparina and Osokin, 2021; Wolfson et al., 2022; Yang et al., 2022; Deng et al., 2022b; Zhao et al., 2022; Niu et al., 2023). In this approach, the natural language question is decomposed into a chain of sub-steps, which has been popular in the context of Knowledge-Graph-based QA with multi-hop questions (Min et al., 2019; Zhang et al., 2019). Recent work attempts to decompose the questions into trees (Huang et al., 2023), which yields explainable answers (Zhang et al., 2023).

In this approach, the question decomposer is sometimes learned in a joint-manner to optimize the performance of an end to end model (Ye et al., 2023); it can also be derived from a syntactic analysis of complex questions (Deng et al., 2022a); or specialized pre-training of decompositions using distant supervision from comparable texts (Zhou et al., 2022); or weak supervision from execution

values (Wolfson et al., 2022). LLMs have also been found effective as generic question decomposers in Chain of Thought (CoT) methods (Wei et al., 2022; Chen et al., 2022; Wang et al., 2023). In this work, we compare our own Question Decomposition method with the QDMR model (Wolfson et al., 2022).

## 3 Decomposing Queries into QPL

### 3.1 Query Plan Language Dataset Conversion

We design *Query Plan Language* (QPL) as a modular dataflow language that encodes the semantics of SQL queries. We take inspiration in our semantic transformation from SQL to QPL from the definition of the execution plans used internally by SQL optimizers, e.g., (Selinger et al., 1979). We automatically convert the original *Spider* dataset into a version that includes QPL expressions for all the training and development parts of *Spider*. The detailed syntax of QPL is shown in § A.4.

QPL is a hierarchical representation for execution plans. It is a tree of operations in which the leaves are *table reading* nodes (Scan nodes), and the inner nodes are either unary operations (such as Aggregate and Filter) or binary operations (such as Join and Intersect). Nodes have arguments, such as the table to scan in a Scan node, or the join predicate of a Join node.

An important distinction between QPL plans and SQL queries is that every QPL sub-plan is a valid executable operator, which returns a stream of data tuples. For example, Fig. 1 shows an execution plan with 4 steps and depth 2. The 4 steps are: the two Scan leaves, the Filter sub-plan, and the Join sub-plan, which is the root of the overall plan.

We automatically convert SQL queries into semantically equivalent QPL plans by reusing the execution plans produced by Microsoft SQL Server 2019 query optimizer (Fritchey, 2018). QPL is a high-level abstraction of the physical execution plan produced (which includes data and index statistics). In QPL syntax, we reduced the number of operators to the 9 operators listed in Table 2. We also design the operators to be *context free*, i.e., all operators take as input streams of tuples and output a stream of tuples, and the output of an operator only depends on its inputs.[3] We experiment with different syntactic realizations of QPL expressions,

---

[3]This is in contrast to SQL execution plan operators such as *Nested-Loops* where the two children nodes tightly depend on each other.

| Operator | Description |
|---|---|
| **Scan** | Scan all rows in a table with optional filtering predicate |
| **Aggregate** | Aggregate a stream of tuples using a grouping criterion into a stream of groups |
| **Filter** | Remove tuples from a stream that do not match a predicate |
| **Sort** | Sort a stream according to a sorting expression |
| **TopSort** | Select the top-K tuples from a stream according to a sorting expression |
| **Join** | Perform a logical join operation between two streams based on a join condition |
| **Except** | Compute the set difference between two streams of tuples |
| **Intersect** | Compute the set intersection between two streams of tuples |
| **Union** | Compute the set union between two streams of tuples |

Table 2: Description of QPL Operators

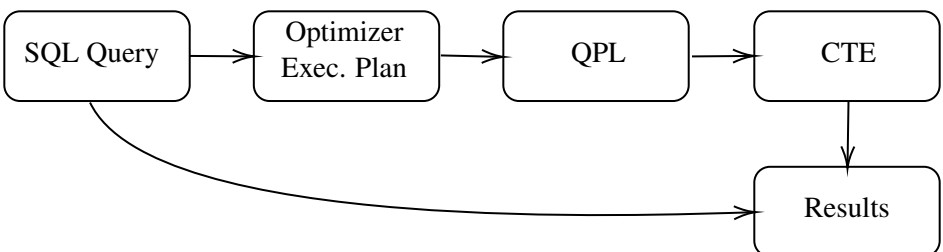

Figure 3: QPL generation process: the dataset SQL expressions are run through the query optimizer, which is then converted into QPL. QPL expressions are converted into modular CTE SQL programs, which can be executed. We verify that the execution results match those of the original SQL queries.

and elected the version where steps are numbered and ordered bottom-up, corresponding roughly to the execution order of the steps. We validate that the result sets returned by the converted QPL plans are equivalent to those of the original Spider SQL queries. We thus enrich the Spider training and development sets with semantically equivalent QPL programs as shown in Fig. 3.

## 3.2 Text-to-QPL Model

**Original SQL**

```
SELECT template_id, count(*)
FROM Documents
GROUP BY template_id
```

## CTE

```
WITH
Scan_1 AS (
    SELECT Template_ID FROM Documents
),
Aggregate_2 AS (
    SELECT COUNT(*) AS Count, Template_ID
    FROM Scan_1
    GROUP BY Template_ID
)
SELECT * FROM Aggregate_2
```

Figure 4: SQL query and its equivalent CTE

In order to train a text-to-QPL model, we fine-tune Flan-T5-XL (Chung et al., 2022) (3B parameters) on 6,509 training examples. Each example contains a question, schema information and the gold computed QPL. The input to the model is the same as in Shaw et al. (2021), *i.e.*, Question | Schema Name | Table1 : Col11, Col12, ... | Table2 : Col21, Col22, ... We also experiment with rich schema encoding, adding type, key and value information as described in §A.2. We train the model for 15 full epochs and choose the model with the best execution accuracy on the development set. Execution accuracy is calculated by generating a QPL prediction, converting it to Common Table Expression format (CTE) (see example in Fig. 4), running the CTE in the database and comparing the predicted result sets of the predicted and gold CTEs. Final evaluation of the model uses the PICARD (Scholak et al., 2021) decoder with a parser we developed for QPL syntax. This constrained decoding method ensures that the generated QPL programs are syntactically valid.

## 4 Question Decomposition

We use the QPL plans automatically computed from SQL queries in the dataset to derive a set of

question decompositions (QD) that are grounded in the QPL steps, as shown in Fig. 1. We investigate three usages of this QD method: (1) $QPL \rightarrow QD$: we learn how to generate a QD given a QPL plan; this is useful at inference time, to present the predicted QPL to non-expert users, as a more readable form of the plan; (2) $Q \rightarrow QD$ we train a question decomposer on the Spider-QPL dataset for which we collect a set of validated automatically generated QDs; (3) $Q + QD \rightarrow QPL$ we finally investigate a Text-to-QPL predictor model which given a question, firstly generates a corresponding QD, and then predicts a QPL plan based on (Q+QD).

## 4.1 QPL to QD

We use the OpenAI `gpt-3.5-turbo-0301` model to generate a QD given a QPL plan. We prepared a few-shot prompt that includes a detailed description of the QPL language syntax and six examples that cover all the QPL operators that we prepared manually (see §A.3).

We validated manually 50 pairs (QPL, QD) generated using this method and found them to be reliable, varied and fluent. In addition, we designed an automatic metric to verify that the generated QDs are well aligned with the source QPL plan: (1) we verify that the number of steps in QD is the same as that in the source QPL; (2) we identify the leaf *Scan* instructions in the QD and verify that they are aligned with the corresponding QPL Scan operations. To this end, we use a fuzzy string matching method to identify the name of the table to be scanned in the QD instruction. The QPL-QD alignment score combines the distance between the length of QD and QPL and the IoU (intersection over union) measure of the set of Scan operations.

## 4.2 Dataset Preparation

Using the QPL $\rightarrow$ QD generator, we further enrich the Spider-QPL dataset with a computed QD field for each sample. For the sake of comparison, we also compute the predicted QDMR decomposition of the question (Wolfson et al., 2020) using the Question Decomposer from (Wolfson et al., 2022)[4]

We obtain for each example a tuple: <Schema, Question, SQL, QPL, QD, QDMR>. We obtained 1,007 valid QDs (QPL-QD alignment score of 1.0) in the Spider Dev Set (out of 1,034) and 6,285 out of 6,509 in the Training Set with a valid QPL.

---

[4]We used the decomposer model published in https://github.com/tomerwolgithub/question-decomposition-to-sql.

## 4.3 Question Decomposer Model

Given the dataset of <Q, QD> obtained above we train a QPL Question Decomposer which learns to predict a QD in our format given a question and a schema description: $Q + Schema \rightarrow QD$. We fine-tune a Flan-T5-XL model for this task, using the same schema encoding as for the $Q + Schema \rightarrow QPL$ model shown in §3.2.

## 4.4 Q+QD to QPL Prediction

We train a Flan-T5-XL model under the same conditions as previous models on $\langle Q, QD, QPL \rangle$ to predict QPL given the QD computed by our question decomposer.

# 5 Experiments and Results

## 5.1 Text-to-QPL Prediction

We present our results on the Spider development set in Table 3. We compare our models to T5-3B with PICARD (Scholak et al., 2021), as it is the closest model to ours in terms of number of parameters, architecture, and decoding strategy. To make the comparison as close as possible, we retrain a model <Q $\rightarrow$ SQL> using the same base model Flan-T5-XL as we use for our <Q $\rightarrow$ QPL> model. We also compare two schema encoding methods: *Simple Schema Encoding* only provides the list of table names and column names for each table; *Rich Schema Encoding* provides for each column additional information: simplified type (same types as used in Spider's dataset - text, number, date, other), keys (primary and foreign keys) and values (see §A.2 for details). We see that at every difficulty level (except "Easy" for Simple Schema Encoding), our Q $\rightarrow$ QPL model improves on the baseline. The same is true compared to the other models in Table 1.[5] All other things being equal, this experiment indicates that it is easier to learn QPL as a target language than SQL **(RQ1)**.

On overall accuracy, the direct Q $\rightarrow$ QPL model achieves a respectable 77.4% without database content and 83.8% with database content. Our model notably achieves highest execution accuracy on Hard and Extra-Hard queries across existing fine-tuned and LLM-based models (70.0% across Hard+Extra-Hard with database content).

---

[5]The Q $\rightarrow$ SQL baseline we show reaches 82% vs. 79% reported in Scholak et al. (2021) as we used a more complete schema encoding and the instruction fine-tuned Flan-T5-XL model as opposed to the base T5-3B model.

| Spider Difficulty | Q → QPL | Q+QD → QPL | Q → SQL | Support |
|---|---|---|---|---|
| **Simple Schema Encoding** | | | | |
| Easy | 87.5% | 84.7% | 91.9% | 248 |
| Medium | 84.3% | 72.2% | 76.9% | 446 |
| Hard | 66.7% | 62.0% | 64.9% | 174 |
| Extra Hard | 54.8% | 45.1% | 44.6% | 166 |
| Overall | **77.4%** | 69.1% | 73.3% | 1034 |
| **Rich Schema Encoding** | | | | |
| Easy | 93.5% | | 91.5% | 248 |
| Medium | 89.0% | | 88.3% | 446 |
| Hard | 74.7% | | 69.5% | 174 |
| Extra Hard | 65.1% | | 63.9% | 166 |
| Overall | **83.8%** | | 82.0% | 1034 |

Table 3: Accuracy on Spider Development Set by difficulty level with *Simple Schema Encoding* (table names and column names) and *Rich Schema Encoding* (column types, keys, values).

| Type | Gold label | Time | Correct |
|---|---|---|---|
| QPL | Incorrect query | 89% | 53% |
| | Correct query | 100% | 79% |
| | | | **67%** |
| SQL | Incorrect query | 102% | 50% |
| | Correct query | 101% | 25% |
| | | | **34%** |
| Avg time | QPL | 123s | |
| | SQL | 132s | |

Table 4: User experiment: 20 (question, query) pairs are shown to 4 users - half in QPL and half in SQL, half are correct and half incorrect. The table reports how long users assessed on average each query, and how often they were right in assessing the correctness of the query.

The <Q + QD → QPL> model is inferior to the direct Q → QPL model (69.1% to 77.4% with simple schema encoding). We verified that this is due to the lower quality of the QD produced by our question decomposer. On Oracle QD, the model produces about 83% accuracy without database content. Table 8 confirms that the model accuracy level increases when the QD-QPL alignment score increases. This indicates that the development of a more robust question decomposer grounded in query decomposition for training signal has the potential to improve semantic parsing performance.

In addition, we find that the <Q+QD → QPL> model produces correct answers that were not computed by the direct <Q → QPL> model in 50 cases

(6 easy, 18 medium, 15 hard, 11 extra hard). This diversity is interesting, because for hard and extra-hard cases, execution accuracy remains low (55%-74%). Showing multiple candidates from different models may be a viable strategy to indicate the lack of confidence the model has on these queries.

## 5.2 Interpretability User Experiment

In order to probe whether QPL is easier to interpret by non-expert SQL users, we organized a user experiment. We selected a sample of 22 queries of complexity Hard and Extra-Hard. We collected predicted SQL queries and QPL plans for the queries, with half correct (producing the expected output), and half incorrect.

Four volunteer software engineers with over five years of experience participated in the experiment. We asked them to determine whether a query in either SQL or QPL corresponded to the intent of the natural language question. The participants were each exposed to half cases in QPL and half in SQL, half correct and half incorrect. We measured the time it took for each participant to make a decision for each case. Results are reported in Table 4. They indicate that the participants were correct about QPL in 67% of the cases vs. 34% of the SQL cases, supporting the hypothesis that validating the alignment between a question and query is easier with QPL than with SQL ($p < 0.06$) (**RQ2**).

| Error Type | Error | Err.% | Explanation |
|---|---|---|---|
| Wrong aggregate | 35 | 21% | Error in Aggregate (sum, avg, count, max, min) |
| Join | 31 | 19% | Wrong join (e.g., not on Primary/Foreign key) |
| Wrong column | 17 | 12% | Output does not include the right columns |
| Missing filter | 17 | 10% | Filter stage is missing |
| Wrong constant | 15 | 9% | Compare with wrong constant (e.g., isOfficial = 'Y' vs. 'T') |
| Wrong predicate | 12 | 7% | Error in selection predicate (e.g., > instead of <) |
| Lost | 12 | 7% | Predicted QPL is completely wrong |
| Typing | 7 | 4% | Compare with constant of wrong type (e.g., age = 'old' vs. 20) |
| Extremum | 4 | 2% | Error in selecting top value (min vs. max for example) |
| Intersect | 4 | 2% | Error in Intersect operation |
| Wrong structure | 3 | 2% | QPL plan is not a connected tree |
| Syntax issue | 3 | 2% | Predicted QPL is not syntactically valid |
| Except | 2 | 1% | Error in Except operation |
| Distinct | 1 | 1% | Missing distinct flag |
| Wrong table | 1 | 1% | Refers to the wrong table in the schema |
| **Grand Total** | **167** | | |

Table 5: Error Types: Breakdown of errors by error types

## 6 Error Analysis

The most challenging error type is related to queries that involve aggregate operations (*group by* and aggregated values such as count, sum or max).

Table 5 shows the breakdown of errors by error types for the Q → QPL model with Rich Schema Encoding. We hypothesize that errors related to Join operations could be reduced by exploiting a more expressive description of the schema structure and applying either a post-processing critique of the generated QPL or enforcing stricter constraints in the QPL Picard parser. Similarly, a more detailed description of the content of encoded columns could improve the *Wrong Constant* type.

The Spider development set includes 20 different schemas. Table 9 shows the break down of errors per schema. We observe that 5 schemas account for over 70% of the errors. These schemas do not follow best practices in data modeling: keys are not declared, column naming is inconsistent, and value encoding in some columns is non standard (e.g., use of 'T'/'F' for boolean values).

## 7 Conclusion

We presented a method to improve compositional learning of complex text-to-SQL models based on QPL, a new executable and modular intermediary language that is derived from SQL through semantic transformation. We provide software tools to automatically translate SQL queries into QPL and to execute QPL plans by translating them to CTE SQL statements. We also compiled Spider-QPL, a version of Spider which includes QPL and Question Decomposition for all examples.

Our experiments indicate that QPL is **easier to learn** using fine-tuned LLMs (our text-to-QPL model achieves SOTA results on fine-tuned models on Spider dev set without db values, especially on hard and extra-hard queries); and **easier to interpret** by users than SQL for complex queries. On the basis of the computed QPL plans, we derived a new form of Question Decomposition and trained a question decomposer that is sensitive to the target database schema, in contrast to existing generic question decomposers. Given a predicted QPL plan, we can derive a readable QD that increases the interpretability of the predicted plan.

In future work, we plan to further exploit the modularity of QPL plans for data augmentation and to explore multi-step inference techniques. We have started experimenting with an auto-regressive model which predicts QPL plans line by line. Our error analysis indicates that enhancing the model to further take advantage of the database values and foreign-keys has the potential to increase the robustness of this multi-step approach. We are also exploring whether users can provide interactive feedback on predicted QD as a way to guide QPL prediction.

## Limitations

All models mentioned in this paper were trained on one NVIDIA H100 GPU with 80GB RAM for 10 epochs, totaling around 8 hours of training per model at the cost of US$2 per GPU hour.

The models were tested on the Spider development set, which only has QPLs of up to 13 lines; our method has not been tested on longer QPLs.

During training, we did not use schema information such as the primary-foreign key relationships and column types, nor did we use the actual databases' content. In this regard, our models might output incorrect `Join` predicates (due to lack of primary-foreign key information), or incorrect `Scan` predicates (due to lack of database content).

We acknowledge certain limitations arising from the evaluation conducted on the Spider development set. These limitations include:

1. A total of 49 queries yield an empty result set on their corresponding databases. Consequently, inaccurately predicted QPLs could generate the same "result" as the gold query while bearing significant semantic differences. In the Spider-QPL dataset, we have inserted additional data so that none of the queries return an empty result set in the development set.

2. As many as 187 queries employ the `LIMIT` function. This can lead to complications in the presence of "ties"; for instance, when considering the query `SELECT grade FROM students ORDER BY grade DESC LIMIT 1`, the returned row becomes arbitrary if more than one student shares the highest grade.

## Ethics Statement

The use of text-to-code applications inherently carries risks, especially when users are unable to confirm the accuracy of the generated code. This issue is particularly pronounced for intricate code, such as complex SQL queries. To mitigate this risk, we introduce a more transparent target language. However, our limited-scale user study reveals that, even when utilizing this more interpretable language, software engineers struggle to detect misaligned queries in more than 30% of instances. This occurs even for queries with moderate complexity (QPL length of 5 to 7). More work on interpretability of generated code is warranted before deploying such tools.

## Acknowledgements

We thank the Frankel Center for Computer Science at Ben Gurion University for their support.

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

## A  Appendix

### A.1  Q → QPL Model

The Q → QPL model was trained using the HuggingFace Transformers (Wolf et al., 2020) and PEFT (Sourab Mangrulkar, 2022) libraries. As the Flan-T5-XL model was too large to fine-tune on one GPU, we use the LoRA (Hu et al., 2021) method to fine-tune only the q and v matrices of the model. Hyperparameters used in training are listed in Table 7. The trained model and training and validation datasets are available on `https://huggingface.co/bgunlp`.

| Hyperparameter | Value |
|---|---|
| # Epoch | 10 |
| Dropout Prob. | 0.05 |
| Batch Size | 1 |
| Learning Rate | 0.0002 |
| Adaptation ($r$) | 16 |
| LoRA $\alpha$ | 32 |

Table 7: Q → QPL Training Hyperparameters

### A.2  Schema Encoding Methods

We compare two methods to describe the database schema when we prompt our models:

1. Simple Schema Encoding: this is similar to Shaw et al. (2021), *i.e.*, `Question | Schema Name | Table1 : Col11, Col12, ... | Table2 : Col21, Col22, ...`

2. Rich Schema Encoding: this encoding provides for each column a simplified type (as in Spider - text, number, date or other); primary and foreign keys; and values.

| QPL Length | Q → QPL | Q+QD → QPL | Support |
|---|---|---|---|
| 1 | 87.3% | 78.3% | 189 |
| 2 | 86.6% | 83.4% | 277 |
| 3 | 85.3% | 78.0% | 191 |
| 4 | 75.0% | 62.9% | 124 |
| 5 | 67.1% | 54.4% | 164 |
| 6 | 48.2% | 25.9% | 27 |
| 7 | 31.8% | 22.7% | 44 |
| ≥8 | 11.9% | 24.4% | 18 |
| Overall | **77.4%** | 69.1% | 1034 |

Table 6: Execution Accuracy of Text-to-QPL Models on *Spider* Development Set by Length of QPL. QPL length is a more natural measure of query complexity than the method used to classify queries in *Spider*. We find that there is little correlation between QPL Length and the Spider difficulty level.

For example:
**Simple Schema Encoding: pets_1**

```
Table Student (StuID, LName, Fname,
    Age, Sex, Major, Advisor, city_code)
Table Pets (PetID, PetType, pet_age,
    weight)
Table Has_Pet (StuID, PetID)
```

**Rich Schema Encoding: pets_1**

```
CREATE TABLE Student (
        StuID number,
        LName text,
        Fname text,
        Age number,
        Sex text,
        Major number,
        Advisor number,
        city_code text,
        primary key ( StuID ))

CREATE TABLE Pets (
        PetID number,
        PetType text ( dog ),
        pet_age number,
        weight number,
        primary key ( PetID ))

CREATE TABLE Has_Pet (
        StuID number,
        PetID number,
        foreign key ( StuID )
          references Student ( StuID ),
        foreign key ( PetID )
          references Pets ( PetID ))
```

Values are added after each column when an n-gram from the question is found as one of the values in one of the rows of the table. For example, for the question *"How much does the youngest dog weigh?"*, the n-gram *"dog"* is found in the values of the column PetType. In this case, the value annotation PetType text ( dog ) is encoded.

### A.3 Question Decomposer Model

The prompt given to ChatGPT (gpt-3.5-turbo) to decompose a question given QPL is listed in Fig. 5. The full prompt (including BNF and all 6 examples) is available on GitHub.

### A.4 QPL Syntax

We show the BNF of the QPL language we have designed in Fig. 6. This grammar is used as part of the PICARD parser used for the decoder of all QPL predictor models.

### A.5 Errors by Schema

Table 9 shows error rate for the Q → QPL model with Rich Schema Encoding by schema for each of the 20 schemas present in Spider's development set.

We observe that 5 of the 20 schemas (car_1, flight_2, student_transcripts_tracking, world_1 and wta_1) account for 41.8% of the examples in the development set and 70.1% of the errors. Most of the errors in these specific schemas can be traced to missing key declarations, inconsistent column naming and typing (strings used to encode boolean or number values).

QPL is a formalism used to describe data retrieval operations over an SQL schema in a modular manner.
A QPL plan is a sequence of instructions for querying tabular data to answer a
natural language question.
Forget everything you know about SQL, only use the following explanations.

A schema is specified as a list of <table> specification in the format:
<table>: <comma separated list of columns>

A plan contains a sequence of operations.
All operations return a stream of tuples.
All operations take as input either a physical table from the schema (for the Scan operation)
or the output of other operations.

Let's think step by step to convert QPL plan to natural language plan given schema,
question, and QPL that describe the question.

In the natural language plan:
1. You must have exactly the same number of questions as there are steps in the QPL.
2. The questions you generate must follow exactly the same order as the steps in the QPL.

This is the formal specification for each operation:

(BNF goes here, elided for brevity)

Example 1:

Schema:
Table Visitor (ID, Name, Age, Level_of_membership)
Table Museum (Museum_ID, Name, Open_Year, Num_of_staff)
Table Visit (Visitor_ID, Museum_ID, Total_Spent, Num_of_Ticket)

Question:
What is the total ticket expense of the visitors whose membership level is 1?

QPL Plan:
#1 = Scan Table [ visitor ] Predicate [ Level_of_membership = 1 ] Output [ ID ]
#2 = Scan Table [ visit ] Output [ visitor_ID , Total_spent ]
#3 = Join [ #1, #2 ] Predicate [ #1.ID = #2.visitor_ID ] Output [ #2.Total_spent ]
#4 = Aggregate [ #3 ] Output [ SUM(Total_spent) AS Sum_Total_spent ]

Natural Language Plan:
#1 = Scan the table Visitor to find who are the visitors with membership level 1
#2 = Scan the table Visit to find what is the total spent by visitors during their visits
#3 = Join #1 and #2 to find what is the total spent by each visitor with
     membership level 1 during their visits
#4 = Group #3 by Visitor and aggregate the sum of total spent to find what is the total spent
     by all visitors with membership level 1 during their visit

(5 more examples are given)

Now your turn:

Schema:
{schema}

Question:
{question}

QPL Plan:
{qpl}

Natural Language Plan:

Figure 5: Prompt used to generate QD

```
<qpl> ::= <line>+
<line> ::= #<integer> = <operator>
<operator> ::= <scan>
             | <aggregate>
             | <filter>
             | <sort>
             | <topsort>
             | <join>
             | <except>
             | <intersect>
             | <union>

-- Leaf operator
<scan> ::= Scan Table [ <table-name> ] <pred>? <distinct>? <output-non-qualif>

-- Unary operators
<aggregate> ::= Aggregate [ <input> ] <group-by>? <output-non-qualif>
<filter>    ::= Filter [ <input> ] <pred> <distinct>? <output-non-qualif>
<sort>      ::= Sort [ <input> ] <order-by> <withTie>? <output-non-qualif>
<topsort>   ::= TopSort [ <input> ] Rows [ <number> ] <order-by>
                        <withTies>? <output-non-qualif>

-- Binary operators
<join>      ::= Join [ <input> , <input> ] <pred>? <distinct>? <output-qualif>
<except>    ::= Except [ <input> , <input> ] <pred> <output-qualif>
<intersect> ::= Intersect [ <input> , <input> ] <pred>? <output-qualif>
<union>     ::= Union [ <input> , <input> ] <output-qualif>

<group-by>  ::= GroupBy [ <column-name> (, <column-name>)* ]
<order-by>  ::= OrderBy [ <column-name> <direction> (, <column-name> <direction>)* ]
<withTies>  ::= WithTies [ true | false ]
<direction> ::= ASC | DESC
<pred>      ::= Predicate [ <comparison> (AND | OR <comparison>)* ]
<distinct>  ::= Distinct [ true | false ]
<output-non-qualif> ::= Output [ <column-name> (, <column-name>)* ]
<output-qualif> ::= Output [ <qualif-column-name> (, <qualif-column-name>)* ]
<qualif-column-name> ::= # <number> . <column-name>
```

Figure 6: QPL Grammar

| QD-QPL Alignment | Support | Correct | Exec Acc | Avg QPL Gold Len. |
|---|---|---|---|---|
| [0.0, 0.4] | 1 | 0 | 0 | 5.0 |
| (0.4, 0.5] | 19 | 4 | 21.1 | 4.6 |
| (0.5, 0.6] | 9 | 2 | 22.2 | 6.3 |
| (0.6, 0.7] | 43 | 9 | 20.9 | 5.0 |
| (0.7, 0.8] | 63 | 21 | 33.3 | 4.2 |
| (0.8, 0.9] | 93 | 33 | 35.5 | 4.6 |
| (0.9, 1] | 779 | 624 | 80.1 | 2.8 |

Table 8: Q+QD → QPL model trained with QDs predicted by Trained Question Decomposer

| Schema ID | Errors | Samples | Error Rate |
|---|---|---|---|
| battle_death | 1 | 16 | 6% |
| **car_1** | **27** | 92 | 29% |
| concert_singer | 5 | 45 | 11% |
| course_teach | 0 | 30 | 0% |
| cre_Doc_Template_Mgt | 8 | 84 | 10% |
| dog_kennels | 12 | 82 | 15% |
| employee_hire_evaluation | 1 | 38 | 3% |
| **flight_2** | **15** | 80 | 19% |
| museum_visit | 2 | 18 | 11% |
| network_1 | 9 | 56 | 16% |
| orchestra | 0 | 40 | 0% |
| pets_1 | 2 | 42 | 5% |
| poker_player | 0 | 40 | 0% |
| real_estate_properties | 1 | 4 | 25% |
| singer | 1 | 30 | 3% |
| **student_transcripts_tracking** | **16** | 78 | 21% |
| tvshow | 5 | 62 | 8% |
| voter_1 | 3 | 15 | 20% |
| **world_1** | **44** | 120 | 37% |
| **wta_1** | **15** | 62 | 24% |
| **Grand Total** | **167** | **1034** | **16%** |

Table 9: Breakdown of errors by Schema ID: 5 schemas out of the 20 present in Spider's development set account for 70% of the errors. These schemas do not follow best practices in data modeling and lack proper foreign key declarations.