# OpenReview forum: "Semantic Decomposition of Question and SQL for Text-to-SQL Parsing"
_EMNLP/2023/Conference — EMNLP 2023 Findings_

### Official Review · Reviewer_zf7K · 2023-08-04

**Soundness:** 3

**Excitement:**

3: Ambivalent: It has merits (e.g., it reports state-of-the-art results, the idea is nice), but there are key weaknesses (e.g., it describes incremental work), and it can significantly benefit from another round of revision. However, I won't object to accepting it if my co-reviewers champion it.

**Missing References:**

N/A

**Paper Topic And Main Contributions:**

This paper proposes Query Plan Language (QPL), a new meaning representation for text-to-SQL parsing that can be adapted for decomposing natural language questions (QD). The paper claims that QPL is easier for models to learn and for non-expert users to understand.

**Questions For The Authors:**

A. While a query optimizer is used to convert SQL queries into execution plans, it is not mentioned how these plans are converted to the proposed QPL language. Other than that, it is also unclear how QPL is translated to CTE and make them executable?

B. How do you validate QPLs have the same execution results as the original SQL queries? Is it a logical proof or empirical test? If the latter, what is the data you use to validate, and what is the coverage of QPL?

C. Why are there only 6,509 training examples? Spider has 7,000 training examples.

**Reasons To Accept:**

This paper attempts to address a salient research question in semantic parsing: meaning representation (MR). The proposed QPL is intended to serve as an MR that reduces model learning difficulty and help non-expert users to identify mistakes in complex SQL queries. The idea of decomposing natural language questions using QPL is interesting, although not sufficiently justified.

**Reasons To Reject:**

I have three major concerns for this paper:
1. The experiments setup cannot sufficiently show that QPL is an effective MR. (1) The authors uses FlanT5 for their methods while using T5 for the baseline T5+PICARD method. It is hard to tell whether the performance gain is brought by QPL or instruction-tuning in FlanT5 itself. (2) The authors does not compare the proposed method with existing work in the main experiment (Table 3), such as  QPL vs NatSQL and QPL -> QD vs QDMR. It is unclear if the proposed methods are better or as effective as existing ones. (3) The experiments are all on the Spider dataset. Since MRs should be generally applicable to the text-to-SQL task, it would be very helpful if the authors can include comparison with open-sourced methods on another text-to-SQL dataset, such as KaggleDBQA (Lee et al., 2021), EHRSQL (Lee et al., 2023), and BIRD (Li et al., 2023).
2. In terms of the user study, the authors draw their conclusion based on p < 0.06, while in such tests p < 0.05 is generally considered statistically significant.
3. This paper lacks analysis on the experiment results. It would be very helpful if the author can provide more insight into why QPL is easier to learn and interpret, especially when compared with existing methods.

This paper lacks elaboration on many important details. Please see the questions listed below.

**Reproducibility:**

3: Could reproduce the results with some difficulty. The settings of parameters are underspecified or subjectively determined; the training/evaluation data are not widely available.

**Reviewer Confidence:**

4: Quite sure. I tried to check the important points carefully. It's unlikely, though conceivable, that I missed something that should affect my ratings.

**Typos Grammar Style And Presentation Improvements:**

While the paper presents a comprehensive review of related work, he first two sections (introduction and related work) use more than half of the main text (4.5 pages), leaving limited space for the authors to introduce their methodology and discuss their experiments in detail. This lack of details hinders the soundness and reproducibility of the paper.

---

> ### Author Rebuttal · Authors · 2023-08-24
>
> **Answers to Questions:**
>
> **A. While a query optimizer is used to convert SQL queries into execution plans, it is not mentioned how these plans are converted to the proposed QPL language. Other than that, it is also unclear how QPL is translated to CTE and make them executable?**
>
> The process of translation "execution plan" -> QPL and QPL -> CTE are standard "programming language" transformations. The key design decision in this process are those listed in the article - which consist of abstracting away physical and index parameters from the execution plan and making sure QPL is "context free" for those operators (such as "nested loops") which are not context-free in the SQL Server execution plan language. As for executing QPL, the key decision is to rely on the extremely useful CTE feature of SQL.
>
> * The translation from SQL Server execution plans (obtained as XML documents) into QPL is described in lines 364-371. The process of translation is implemented as a Scala program - with a parser of the XML plans (200 lines of type definitions and 400 lines of parsing code) and an AST to AST recursive translator (800 lines of Scala code covering all types of operators in the SQL Server execution plan syntax).
> * The translation from QPL to CTE is implemented in Python (400 lines) - it produces a sequence of simple SQL queries that are bound together into a single CTE expression.
>
> Our github provides this code as well as notebooks and interactive Web User Interface tools to experiment with these transformations on arbitrary queries.
>
> **B. How do you validate QPLs have the same execution results as the original SQL queries? Is it a logical proof or empirical test? If the latter, what is the data you use to validate, and what is the coverage of QPL?**
>
> The pipeline consists of translating original SQL queries to execution plans (we rely on the correctness of the SQL optimizer at this stage), then translate each step in the execution plans into QPL queries which are simplified syntactic realizations of simple SQL constructs (one for each QPL primitive operator), which are then bound into a CTE expression.
>
> We validated empirically that this transformation preserved execution results on the Spider dataset (as well as on augmented tests which are not reported in the paper) - one the 6,509 training samples that we successfully converted from SQLite to T-SQL (see question C) and the 1,034 samples from the dev set.  In some cases, the Spider databases contain little data and some queries return empty result sets.  In these cases, we augmented the content of the databases so that all queries return non-empty result sets and we confirmed that the result sets are equivalent.
>
> C. Why are there only 6,509 training examples? Spider has 7,000 training examples.
>
> The results we report are based on an automatic translation pipeline from the original Spider dataset in SQLite to QPL.
> This translation pipeline failed on the missing 491 samples for a variety of reasons:
>
> * The original SQLite query could not be translated into a valid SQL Server query because of the "non strict" nature of SQLite (no type checking, no enforcement of aggregation rules, loose implementation of referential integrity)
> * The generated execution plan included a rare configuration that we did not cover as part of our transformation logic.
>
> We found that the identification of these problematic SQLite cases through translation to a strict T-SQL variant is a valuable contribution to better analyze Spider results - and we will add information about this process in Appendix.
>
> **Reaction to Reason to Reject:**
>
> **R1. The experiments setup cannot sufficiently show that QPL is an effective MR.**
> **(1) The authors uses FlanT5 for their methods while using T5 for the baseline T5+PICARD method. It is hard to tell whether the performance gain is brought by QPL or instruction-tuning in FlanT5 itself.**
>
> We actually did use the same FlanT5-XL model for finetuning the Q->SQL with Picard as we used for QPL - thank you for noting this discrepancy.  We will clarify it in the paper.  This experiment is intended to compare SQL with QPL as a target language "all other things being equal".
>
> **(2) The authors do not compare the proposed method with existing work in the main experiment (Table 3),
> such as QPL vs NatSQL and QPL -> QD vs QDMR. It is unclear if the proposed methods are better or as effective as existing ones.**
>
> We do report baselines for QPL vs. NatSQL in Table 1 - as well as other recent LLM-based few-shot methods (Din-SQL GPT4 as well as GPT-3.5) under similar experimental settings as our experiments (Spider Dev Set split by difficulty level).  The experiment we report in Table 3 is a targeted experiment where we specifically compare SQL vs. QPL as a target language "all other things being equal" - using the same constrained decoding method (Picard adopted to QPL with an incremental QPL parser we developed).
>
> QDMR is not a query language - it is used as a set in a question decomposition approach. We did compare (Q+QDMR->QPL) vs (Q+QD=>QPL) and obtained lower results than with (Q+QD) - we did not report this experiment as it did not bring direct clarity on the contribution, compared to the discussion of Oracle QD we report in lines 476-483.
>
> **(3) The experiments are all on the Spider dataset. Since MRs should be generally applicable to the text-to-SQL task,
> it would be very helpful if the authors can include comparison with open-sourced methods on another text-to-SQL dataset,
> such as KaggleDBQA (Lee et al., 2021), EHRSQL (Lee et al., 2023), and BIRD (Li et al., 2023).**
>
> In principle, it is indeed better to test on more databases - but it is not clear how spending resources on testing on more datasets would change the overall contribution of this work. KaggleDBQA is a nice dataset, but it has had relatively little impact on the community (few citations compared to Spider and few comparisons with it across different methods); it also focuses on demonstrating the contribution of database documentation to few-shot LLM-based semantic parsers - which is interesting but orthogonal to the aspect of query complexity we discuss in this contribution; EHRSQL is a single schema dataset (compared to ~150+ different schemas in different domains in Spider dev+training sets); BIRD is a very recent contribution (the paper is dated May 2023) and focuses on using values in the databases (which we decided not to do in our first experiments to compare with previous work on Spider).
>
> **R2. In terms of the user study, the authors draw their conclusion based on p < 0.06, while in such tests p < 0.05 is generally considered statistically significant.**
>
> Indeed 0.05 is the standard - yet, 0.06 is not a baseless observation either (see <https://www.ncbi.nlm.nih.gov/pmc/articles/PMC4111019/> for a discussion of p-value fallacies).
>
> **R3. This paper lacks analysis on the experiment results. It would be very helpful if the author can provide more insight into why QPL is
> easier to learn and interpret, especially when compared with existing methods.**
>
> We discuss this point in our motivation section (lines 273-294) in our comparison with existing intermediary representation methods (mainly NatSQL) - learning a more compositional and regular representation of queries allows models to generalize better to more complex queries. More experiments are necessary to "explain" why a modular query language is easier to interpret and learn than a non-modular one - but contributing the QPL language, the pipeline to produce an experiment on ~7,500 (query, question) pairs and presenting supporting evidence that modularity helps in learning and interpretation is a first step in obtaining such explanations in the future.
>
> **Presentation Improvements:**
>
> > The first two sections (introduction and related work) use more than half of the main text (4.5 pages)
>
> These two sections include critical material related to our approach and are not just survey of previous work:
>
> * Figure 1 illustrates our approach (2/3 of page 2)
> * Sections 2.3 and 2.4 compare existing approaches to intermediate representations to our approach and to QD.  These contribute to the presentation of our approach and are not only a survey of previous work.
>
> **Reproducibility: 2**
>
> Our github repository includes:
>
> * Code for a fully automatic pipeline to translate the original Spider sqlite data to T-SQL / gather execution plans / translate execution plans to QPL / convert QPL to CTE / validate that the result sets obtained are equivalent to those of the original SQL queries for both training and development sets of Spider.
> * A QPL incremental parser used for the adaptation of the Picard constrained decoding method to QPL and to validate QPL plans interactively.
> * Finetuning code to reproduce the experiments reported in the paper.
> * The Spider-QPL dataset in the form of two JSON files including question, original SQL query, QPL, CTE, result sets for both Dev and Training sets and the Question Decomposition (QD) generated by our question decomposer as well as the QDMR question decomposition produced by the QDMR authors project.
>
> As such we believe our contribution meets the criteria for a higher reproducibility assessment.

---

### Official Review · Reviewer_wWqA · 2023-08-04

**Soundness:** 4

**Excitement:**

4: Strong: This paper deepens the understanding of some phenomenon or lowers the barriers to an existing research direction.

**Missing References:**

The related works are adequately cited.

**Paper Topic And Main Contributions:**

The paper presents the Modular Query Plan Language (QPL), offering an alternative approach to traditional SQL for handling complex queries. By breaking down SQL queries into simpler sub-queries, QPL proposes a modular structure that could potentially benefit Large Language Models (LLMs).

A notable strength of QPL is its emphasis on simplicity and modularity. The authors argue that this design could make it easier for LLMs to learn and comprehend complex logic, which is a crucial aspect in natural language processing. With QPL, LLMs might be able to handle intricate queries more effectively, potentially leading to improved performance.

The paper claims that LLMs perform better when using QPL compared to regular SQL, supported by evidence from experiments and results. This suggests that QPL might offer advantages over traditional SQL in handling complex queries and improving model accuracy. Furthermore, the paper highlights the potential interpretability benefits of QPL over SQL. The modular design and straightforward syntax of QPL could make it more user-friendly and easier to understand. This aspect is essential in promoting transparency and understanding.

In addition to proposing QPL, the paper contributes by augmenting the SPIDER dataset with QPL annotations. This augmentation could be a valuable resource for researchers interested in working with modular query languages.

The proposed text-to-QPL model, which utilizes a fine-tuned FlanT5-XL with a constrained decoding algorithm, demonstrates the practical applicability of QPL.

**Questions For The Authors:**

1.  While Q + QD -> QPL is an interesting experiment, why does having additional information hurt as compared to Q -> QPL ?

2. How does In-context learning/few shot learning using large models (as described in Table 1) perform with QPL?

3. Are execution plans the best way of decomposing a SQL query, considering that execution plans are optimized for performance, the paper could analyze whether such optimization aligns perfectly with the interpretability and modularity goals of QPL.?

4. A crucial aspect of QPL is its interpretability. However, it would be valuable to expand on how QPL fares against QDMR concerning interpretability with non-experts. Providing specific examples or user studies to showcase how well non-experts can understand and reason about the queries in each representation could add depth to the paper's discussion on the interpretability aspect of QPL. This comparison could highlight the practical implications and user-friendliness of QPL in real-world scenarios.

**Reasons To Accept:**

1. By breaking down complex SQL queries into simpler sub-queries, QPL showcases its potential to handle challenging tasks more efficiently. The experimental results provide convincing evidence of QPL's ability to improve the performance of language models (LLMs) when dealing with complex queries.

2. With the use of QPL, the paper argues that the interpretability of the queries is enhanced, allowing for better transparency and understanding of LLMs' outputs.

3. The augmentation of the SPIDER dataset with QPL annotations can be a valuable contribution of this paper.

4. The empirical results presented in the paper provide strong support for the claims made about the usefulness of QPL compared to traditional SQL.

**Reasons To Reject:**

1. The paper presents advantages of using the Modular Query Plan Language (QPL) over SQL; however, it lacks a comparison with other decompositions, notably the Query Decomposition Meaning Representation (QDMR), in terms of interpretability. Including such a comparison could provide a more comprehensive understanding of QPL's strengths and weaknesses in relation to existing approaches.

2. The paper would benefit from an error analysis that examines the mistakes made with QPL compared to SQL as a language. This analysis could shed light on specific challenges and issues that arise with QPL, especially considering that it is proposed primarily for the semantic parsing task. Incorporating concrete examples of these mistakes would be helpful for readers to gain deeper insights into the practical implications of using QPL.

**Reproducibility:**

5: Could easily reproduce the results.

**Reviewer Confidence:**

4: Quite sure. I tried to check the important points carefully. It's unlikely, though conceivable, that I missed something that should affect my ratings.

**Typos Grammar Style And Presentation Improvements:**

The paper reads well.

Minor

1. line 462 - use consistent notation for the model (FlanT5-XL).

---

> ### Author Rebuttal · Authors · 2023-08-24
>
> **Answer to questions:**
>
> 1. **While Q + QD -> QPL is an interesting experiment, why does having additional information hurt as compared to Q -> QPL?**
> We were indeed curious about this point. We believe this aspect is addressed in the paragraph 476-482: The QD we use in the experiment is computed by the Question Decomposer model we used. When using Oracle QD - we do observe much higher performance compared to direct Q->QPL. This experiment motivates further research in learning a more accurate Question Decomposer. See Table 7 in Appendix which indicates that a simple metric of QD/QPL alignment can provide guidance to improve the learning of a better Question Decomposer.
> 2. **How does In-context learning/few shot learning using large models (as described in Table 1) perform with QPL?**
> We indeed performed the experiment to compare Text-to-QPL using few-shot LLMs as in Table 1 - the results are lower than on SQL using the same prompting strategies. We have performed analysis to explain this discrepancy - working hypotheses at this point are that:
> - (a) GPT3.5 like LLMs "know" SQL as part of their pre-training which provides strong prior for specifically generating SQL as opposed to "unknown languages" like QPL without further fine-tuning;
> - (b) we suspect the Spider dev dataset is "memorized" by these models to some extent (establishing this point is difficult but we tend to believe proving the opposite should be the burden of proof).
> 3. **Are execution plans the best way of decomposing a SQL query, considering that execution plans are optimized for performance, the paper could analyze whether such optimization aligns perfectly with the interpretability and modularity goals of QPL.?**
> This is a fascinating question - there are indeed different execution plans that can be generated for the same query depending on statistical information gathered by the SQL optimizer. SQL optimizers optimize for performance - and not for "interpretability" - although it is complex to model "interpretability" in this context. Yet, regardless of the specific plan selected by the optimizer, we do obtain a semantic decomposition of the query into a restricted set of simple operators. In other words, the execution plan achieves "modularity" in breaking down complex queries into a plan of simple steps, with a simple uniform semantic of "stream of tuples flow" across the nodes of the plan. In further analysis, we have measured the distribution of "high-level set operations" which we associate with high-level interpretable structures (union, intersection, set difference) and we identified that these are present in the QPL plans we have computed on the basis of the optimizer data with high frequency. Future research should help us compare different planning strategies when decomposing complex queries to identify those with higher interpretability.
> 4. **A crucial aspect of QPL is its interpretability. However, it would be valuable to expand on how QPL fares against QDMR concerning interpretability with non-experts. (...)**
> QPL is not directly comparable to QDMR because QPL is a semantically equivalent transformation of the original SQL queries, that ensures the exact same results at runtime and is database schema aware. QDMR is more comparable with the QD we obtain from QPL. Unlike QPL, QDMR is not deterministically executable, it requires an additional step of SQL synthesis, where the result is an SQL which is at least as complex as the gold SQL, therefore not more interpretable than the gold SQL.
> Overall, the comparison of how QPL / QD / QDMR fare in interpretability is definitely an interesting direction for future research - thank you for the suggestion.
>
> **Reaction to "Reasons to Reject":**
>
> **R1. The paper compares QPL with SQL - but not QPL with QDMR in terms of interpretability.**
>
> There is a design decision whether we aim to decompose the natural language question (Q->QD) or the query (SQL->QPL).
> The two approaches can converge - (SQL->QPL / Q->QDMR). The novelty of our approach in the context of Text-to-SQL is to propose a *semantic* method to decompose complex queries into structured plans. This is in contrast to *syntactic* simplifications of SQL (as in NatSQL). It is also in contrast with *heuristic* question decomposition strategies (as in QDMR) which originate from multi-hop QA.
>
> In terms of interpretability, as mentioned above, QPL is not directly comparable to QDMR because QPL is an executable programming language. Future research should cover user studies comparing SQL / QPL / QD / QDMR with people at different expertise level in SQL as well as in the database domains (our error analysis indicates people as well as model fail because of lack of domain knowledge).
>
> **R2. Lack of error analysis on predicted QPLs**
>
> We agree such error analysis will improve the paper - and will be happy to provide it in the paper.
> It is worth noting that in our experiment we compared SQL to QPL under "basic conditions" - that is, without taking into account
> values in the database tables and exploiting column types and detailed keys relations. Error analysis indicates that these parameters
> as well as domain specific schema linking knowledge become key systemic contributors to QPL errors.  Our latest experiments confirm proper modeling of these schema parameters plays an important role in QPL prediction.

---

### Official Review · Reviewer_fkDx · 2023-08-05

**Soundness:** 3

**Excitement:**

3: Ambivalent: It has merits (e.g., it reports state-of-the-art results, the idea is nice), but there are key weaknesses (e.g., it describes incremental work), and it can significantly benefit from another round of revision. However, I won't object to accepting it if my co-reviewers champion it.

**Paper Topic And Main Contributions:**

The paper proposes a modular Query Plan Language (QPL) to decompose SQL queries into regular sub-queries. The authors develop a translator from SQL to QPL and augment the Spider dataset with QPL programs. The experimental results suggest that text-to-QPL parsing is more effective than text-to-SQL parsing for equivalent queries, and QPL offers benefits in question decomposition and interpretability.

**Reasons To Accept:**

1. The novel QPL offers a systematic approach to decomposing complex SQL queries into manageable sub-queries. The development of a translator leveraging SQL server query optimization plans provides a practical tool for converting SQL queries into QPL, enhancing the applicability of the approach.
2. Experimental results support the superiority of text-to-QPL parsing over text-to-SQL parsing for equivalent queries, reinforcing the practical significance of the proposed approach.
3. QPL's ability to be paraphrased as simple questions and its accessibility to non-experts extend its utility beyond semantic parsing, adding interpretability and broader application potential.

**Reasons To Reject:**

1. While the paper highlights the advantages of the proposed Query Plan Language (QPL), it appears to lack comprehensive experimental results that robustly validate these advantages. More detailed experiments, including comparisons with existing methods, varied datasets, and a thorough statistical analysis, would provide stronger evidence for the claimed benefits of QPL.
2. The application scenario for the intermediate representation created by QPL is not well-defined or explored in the paper.  Addressing this aspect would enhance the paper's relevance to practitioners and potential adopters of the approach.

**Reproducibility:**

4: Could mostly reproduce the results, but there may be some variation because of sample variance or minor variations in their interpretation of the protocol or method.

**Reviewer Confidence:**

3: Pretty sure, but there's a chance I missed something. Although I have a good feel for this area in general, I did not carefully check the paper's details, e.g., the math, experimental design, or novelty.

---

> ### Author Rebuttal · Authors · 2023-08-24
>
> **Reaction to "Reasons to Reject":**
>
> *R1: Lack experimental results to validate QPL's advantages.*
>
> We presented a series of experimental results:
>
> 1. Similar methods (fine-tuned 3B params LLM with Picard constrained decoding) provide much better results on Hard/Extra-hard queries (+5% overal, +10% for extra-hard specifically).  This is a targeted experiment comparing just the effect of changing the meaning representation from SQL to modular QPL.
> 2. In Table 1 - we also compare with a variety of different recent and influential methods on the Spider dataset - NatSQL+RATSQL, Din-SQL GPT4 and a reproduction of recent few-shot GPT3.5 experiments - and observed benefits on complex queries as well.
> 3. We also evaluated the capability of experimented programmers to detect whether a predicted query corresponds to a question intent - and showed a significant advantage for QPL over complex SQL.
> 4. We demonstrated the capability to train a question decomposer which decomposes a natural language question into simpler steps grounded on the QPL operators.
>
> Taking into account that the paper presents both a method to design the new language (QPL), generate the new formalism
> and the curation of the dataset and training of initial models demonstrating the practical advantage of predicting QPL vs. SQL
> we believe the contribution for a single paper is significant.  We are in the process of investigating more advantages of QPL - but
> we still believe that the definition of the formalism, the practical availability of the pipeline producing QPL from SQL and of
> the Spider-QPL dataset provide value for researchers.
>
> *Reason to reject 2: application scenario for QPL is not well-defined or explored in the paper.*
>
> This argument is quite general - it is not exactly clear what you mean by "application scenario" - assuming we mean "using QPL as a query representation instead of SQL for researching semantic parsing and issues related to compositional generalization" - then we believe that the "Reasons to Accept" you have listed somehow answer this point. Our motivation is to design an intermediary language that is (1) semantically equivalent to original complex SQL queries, (2) executable, (3) easy to interpret by programmers, (4) easier to predict by ML compositional models. By making available the code for producing Spider-QPL from the existing Spider dataset, we constructively support objectives 1 and 2. The experimental results we report support 3 (with the finetuning Text-to-QPL experiment) and 4 (with the human analysis and the generation of decomposed questions in natural language grounded in QPL). We also believe that the structure of QPL allows more sophisticated training methods than the end-to-end fine-tuning baseline we presented - including methods exploiting question decomposition and auto-regressive training of QPL statements.
>
> Beyond semantic parsing research, our small user study indicates there is potential for using QPL in a more general "application scenario". As you noted, we believe there is more potential in using QPL for more future research around interpretability - we find the fact that programmers cannot detect the validity of SQL statements in over 50% of the cases extremely alarming - and this requires vigorous future research, especially given the rapidity with which code-assistance tools are being deployed in production.
>
> **Reaction to: Reproducibility: 3**
>
> Our github repository includes:
>
> * Code for a fully automatic pipeline to translate the original Spider sqlite data to T-SQL / gather execution plans / translate execution plans to QPL / convert QPL to CTE / validate that the result sets obtained are equivalent to those of the original SQL queries for both training and development sets of Spider.
> * A QPL incremental parser used for the adaptation of the Picard constrained decoding method to QPL and to validate QPL plans interactively.
> * Finetuning code to reproduce the experiments reported in the paper.
> * The Spider-QPL dataset in the form of two JSON files including question, original SQL query, QPL, CTE, result sets for both Dev and Training sets and the Question Decomposition (QD) generated by our question decomposer as well as the QDMR question decomposition produced by the QDMR authors project.
>
> As such we believe it meets the criteria for a higher reproducibility assessment.

---

### Meta-Review · Area_Chair_tVKv · 2023-09-19

**Recommendation:** 4

**Metareview:**

This paper presents a query plan language to decompose sql queries. The problem is text-to-sql translation / parsing. The method is sound. I find the decomposition strategy to be sound, and it could potentially bring some interpretability down the road. The results demonstrate the model is effective.

The reviewers pointed out a few suggestions to strengthen the paper: running experiments with more (and commonly used) datasets, and more importantly, analysis describing what kind of queries their method is most effective with (which they shared in the discussion, so it would be easy to add it to the camera-ready version).

Not running experiments with papers published in 2023 is not negative, as they are considered contemporaneous. This is minor and easily fixable, but claiming p-value < 0.06 goes against the norm. The results are just not statistically significant (and that is probably fine). Finally, the authors are committing to release their implementation, so it is hard to justify anything but a high score for Reproducibility.

---

### Decision · Program_Chairs · 2023-10-07

**Decision:**

Accept-Findings

**Comment:**

This paper presents a query plan language to decompose sql queries. The problem is text-to-sql translation / parsing. The method is sound. I find the decomposition strategy to be sound, and it could potentially bring some interpretability down the road. The results demonstrate the model is effective.

The reviewers pointed out a few suggestions to strengthen the paper: running experiments with more (and commonly used) datasets, and more importantly, analysis describing what kind of queries their method is most effective with (which they shared in the discussion, so it would be easy to add it to the camera-ready version).

Not running experiments with papers published in 2023 is not negative, as they are considered contemporaneous. This is minor and easily fixable, but claiming p-value < 0.06 goes against the norm. The results are just not statistically significant (and that is probably fine). Finally, the authors are committing to release their implementation, so it is hard to justify anything but a high score for Reproducibility.